# Drug Discovery Based on Fluorine-Containing Glycomimetics

**DOI:** 10.3390/molecules28186641

**Published:** 2023-09-15

**Authors:** Xingxing Wei, Pengyu Wang, Fen Liu, Xinshan Ye, Decai Xiong

**Affiliations:** 1Department of Pharmacy, Changzhi Medical College, No. 161, Jiefang East Street, Changzhi 046012, China; 2State Key Laboratory of Natural and Biomimetic Drugs, School of Pharmaceutical Sciences, Peking University, Xue Yuan Rd. No. 38, Beijing 100191, Chinaliufen@bjmu.edu.cn (F.L.); xinshan@bjmu.edu.cn (X.Y.)

**Keywords:** glycomimetics, fluorination, synthesis, physical properties, drug discovery

## Abstract

Glycomimetics, which are synthetic molecules designed to mimic the structures and functions of natural carbohydrates, have been developed to overcome the limitations associated with natural carbohydrates. The fluorination of carbohydrates has emerged as a promising solution to dramatically enhance the metabolic stability, bioavailability, and protein-binding affinity of natural carbohydrates. In this review, the fluorination methods used to prepare the fluorinated carbohydrates, the effects of fluorination on the physical, chemical, and biological characteristics of natural sugars, and the biological activities of fluorinated sugars are presented.

## 1. Introduction

Carbohydrates play major structural, physical, and biological roles in organisms. They often form complex and diverse glycans on the cell surface by interacting with protein and lipid scaffolds. The glycan molecules bind to proteins and actively participate in various biological processes, such as embryogenesis, adhesion, immunity, inflammation, cancer metastasis, and host–pathogen interactions [1,2,3]. The composition of glycans changes during cell differentiation and tissue development. Disease states and the degree of inflammation also influence the compositional changes. These changes can be attributed to the altered degree of the expression of glycosidase and/or glycosyltransferase (GT) in cells. The generation of this dynamic expression is an integral part of the cell-to-cell communication process, and it can potentially provide new targets for disease treatment. Carbohydrate–protein binding is achieved by exploiting low-energy interactions (such as hydrogen bonding, salt bridges, and metal chelation), which cannot compensate for the high enthalpy cost of the desolvation of polar substrates and shallow protein-binding sites. The high polar surface area of natural carbohydrates hinders the passive penetration of the molecules into the intestinal membrane, making them non-bioavailable orally. It has also been observed that natural carbohydrates exhibit inherently poor pharmacokinetic properties, limiting their therapeutic potential [4,5].

Natural carbohydrates can be modified to improve their drug-like properties, and “carbohydrate-like compounds” can be designed to mimic the structure and function of natural carbohydrates to improve their affinity, pharmacokinetic properties, and bioavailability [6,7]. Fluorine, which exhibits unique properties, has been widely used in drug design and development. Its high electronegativity makes it a powerful tool for modulating the pKa and electron density of the groups present in its proximity, and it can serve as a considerable element to control molecular conformation [8,9,10]. Pharmaceuticals containing fluorine exhibit the property of tunable lipophilicity and increased metabolic stability. It is noteworthy that the oxidative metabolism of these molecules can be prevented. The strategy of the fluorination of carbohydrates has long been used to investigate protein-carbohydrate and carbohydrate-carbohydrate interactions. This method has been used to investigate the contributions of individual sugar alcohol groups [11,12,13] or design mechanism-based inhibitors [14,15]. The excellent NMR (nuclear magnetic resonance) properties of the ^19^F nucleus have been analyzed to study protein-carbohydrate binding at the molecular level [16,17,18,19,20,21,22,23,24]. The method of fluoridation is also used in drug discovery programs and to develop synthetic carbohydrate vaccines [25,26,27,28,29,30]. Fluorination could optimize the physicochemical properties, absorption ability, and distribution properties of natural sugars, and the introduction of fluorine can also help improve the binding affinity and pharmacokinetic properties of natural sugars [31,32]. This review aims to provide up-to-date information on the synthesis method, properties, and biological applications of fluorinated carbohydrates.

## 2. Synthesis of Fluorine-Containing Glycomimetics

In essence, the fluorine-based carbohydrate modification method is followed to substitute various atoms on the molecular skeleton with fluorine atoms or fluorine-containing groups such as polyfluorene. The fluorinating reagents used for fluorination include deoxyfluorination agents: DAST (diethylaminosulfur trifluoride, **1**) [33,34]; nucleophilic fluorination agents: TASF (tris(dimethylamino)sulfonium difluorotrimethylsilicate, **2**) [35]; TBAF (tertbutylammonium fluoride, **3**) [36]; MF_n_ (Metal fluorides, **4**) [37]; KHF_2_ (**5**) [38]; anhydrous hydrogen fluoride system (**6**) [39]; electrophilic fluorination agents: SelectFluor (1-chloromethyl-4-fluoro-1,4-diazoniabicyclo[2.2.2]octane bis(tetrafluoroborate), also abbreviated as F-TEDA-BF_4_, **7**) [40]; NFSI (N-fluorobenzenesulfonimide, **8**) [41]; etc. (Figure 1). With the use of these fluorinating reagents above, the reactions for fluorination of sugars could be divided into nucleophilic reactions, electrophilic addition reactions, radical reactions, metal-catalyzed reactions, and de novo synthesis with fluorine-containing blocks. In this subsection, the synthesis methods will be described in detail according to the reaction types.

### 2.1. Fluorination of Carbohydrates via Nucleophilic Fluorination

A large number of fluorine-containing carbohydrates are synthesized via nucleophilic reactions with fluoride ions. Depending on the reaction condition, fluoride ions can act as good nucleophilic reagents in polar non-protonic solvents [42]. Currently, fluorinated reagents such as DAST (diethylaminosulfur trifluoride), TASF (tris(dimethylamino)sulfonium difluorotrimethylsilicate), KHF_2_, etc., are mainly used to provide nucleophilic fluorine atoms, which subsequently replace the hydroxyl or sulfonate groups on the sugar ring to obtain fluorinated sugars. These are the most popular methods for the preparation of fluorinated carbohydrates [43,44,45,46] (Figure 2a–c). The fluorination of the equatorial hydroxyl group obtained **10** with fluorine nucleophilic substituted in the axial position, which undergoes an S_N_2 reaction mechanism. The stereo-configuration of the fluorination product is generally flipped from the starting material.

While reagents like DAST and TASF have been traditionally used in fluorination reactions, researchers have encountered certain drawbacks associated with their use. For example, protecting group migration can be a significant issue in some fluorination reactions, leading to reduced yields or even the absence of the desired target product [47]. DFMBA is considered milder compared to reagents like DAST and exhibits improved compatibility with various protecting groups [48,49], such as acetyl, methyl, silyl, benzyl, etc. (Figure 2d). In addition to the above-mentioned organic fluorinating agents, other inorganic fluorine-containing compounds, such as hydrogen fluoride, TBAF, CsF, AgF, iodine–fluorine compounds, etc., can also provide nucleophilic fluoride ions [45]. Cheeseman’s group successfully synthesized fluorinated furanose **18** using CsF as the fluorinating agent. This method allowed for the introduction of fluorine-18 into the substrate. After deprotection, the resulting product **19** was found to be suitable as a contrast agent for breast cancer (Figure 2e).

Polyfluorinated carbohydrates have gained attention due to their unique properties and potential applications in medicinal chemistry. The positional combinations for deoxy polyfluorination of pentoses and hexoses have been extensively studied in the context of nucleophilic fluorination reactions. Epoxide opening with fluoride sources is a versatile method for introducing fluorine in a regioselective manner. Thus, 2,3,4-trideoxy-2,3,4-trifluoro-D-glucopyranose (**24**) can be synthesized selectively, utilizing the regioselective opening of the epoxide, displacement of the triflate intermediate using Et_3_N·3HF, and subsequent nucleophilic fluorination with DAST (Figure 3). The synthesis of polyfluorinated carbohydrates, including ketosugars (such as sialic acids), aminosugars, and nucleosides, with controlled fluorine introduction at specific positions has garnered significant attention in recent years. This area of research has been recently reviewed [50].

The nucleophilic reaction of carbonyl-containing glycans with difluoroalkylation reagents produces difluorinated modification of glycans. This type of reaction was first reported by Bobek et al. in 1977 [51]. They utilized a DAST reagent to modify furanose **25** by introducing a difluoromethyl group, resulting in the formation of product **26** (Figure 4a). However, the yield of this reaction was low, and its applicability was limited, particularly for pyranose. In 2009, Jiménez-Barbero et al. [52] successfully synthesized difluorine-modified carboglycosides **28** using the Deoxo-Fluor reagent (Figure 4b). CBr_2_F_2_ is another commonly used reagent for difluorination, often in conjunction with additional phosphorus-containing small molecules to generate the active intermediate carbene. Slawin’s group [53] employed this method to successfully synthesize compound **30** (Figure 4c), which, upon reduction, produced the extracyclic difluoromethyl glycosides **31**.

In recent years, newly developed difluoroalkylation reagents have emerged as effective tools for the difluorinated substitution modification of sugars. One commonly used and cost-effective difluoroalkylation reagent is BrCF_2_CO_2_Et [54]. Quirion’s group [55] utilized the Reformatsky reaction to efficiently and smoothly introduce the CF_2_CO_2_Et moiety into lactone **32**, yielding the exocyclic difluorocarboside **33** (Figure 5a). This reaction exhibits high stereoselectivity and directly forms the β-mannose carboglycoside bond. Another reagent frequently employed for introducing 2-(ethoxycarbonyl)difluoromethyl is difluoroenol silyl ether. Quirion’s group [56], in 2009, employed this reagent in conjunction with boron trifluoride ether as a catalyst to couple with compound **34**, resulting in the formation of exocyclic difluorocarboside **35** (Figure 5b).

The trifluoromethylation of carbohydrates can be achieved via the nucleophilic addition of trifluoromethyl groups to aldehydes or ketones. A commonly employed nucleophilic trifluoromethyl reagent is TMSCF_3_. This reagent enables the direct introduction of trifluoromethyl to the carbonyl carbon in the sugar structure. For instance, TMSCF_3_ was used to treat lactone **36**, leading to the formation of compound **37** (Figure 6a) [57]. Moreover, TMSCF_3_ can also react with other carbonyl-containing sugars and subsequently undergo the Barton–McCombie dehydroxylation reaction to produce the corresponding trifluoromethyl-modified carbohydrates (Figure 6b,c) [58].

### 2.2. Fluorination of Carbohydrates via Electrophilic Addition Reactions

In contrast to nucleophilic reactions, electrophilic addition reactions predominantly involve glycals. The commonly utilized reagents for these reactions are N-F reagents and F-O reagents. N-F reagents are well established, commercially available, and have proven their usefulness in numerous important reactions [59]. Established and readily available N-F reagents have demonstrated their utility in a variety of important reactions [59]. For instance, the Selectfluor reagent has proven to be a highly efficient and mild reagent for fluorination, surpassing the effectiveness of the DAST reagent. In 1997, Wong’s group [60] discovered that the Selectfluor reagent could be used for the one-pot synthesis of 2-fluoroglycosides (Figure 7a). The resulting product, 2-fluoro-2-deoxyglycoside **46**, was utilized as a probe to investigate the catalytic mechanisms of glycosidases and glycosyltransferases. In that same year, Dax’s group [61] successfully achieved electrophilic addition to glycals using another N-F reagent known as N-fluorobenzenesulfonamide (Figure 7b).

In addition to N-F-type fluorination reagents, certain O-F-type reagents can also be employed in the electrophilic addition reactions of glycals. In 1970, Adamson’s group [62] first synthesized 2-Fluoro-2-deoxy-D-glucose **49** using CF_3_OF (Figure 8). Since then, efforts have been made to improve this method. In 2006, Schrobilgen’s group [63] achieved the introduction of a fluorine atom into C-2 of glucal **34** using AcO^18^F (Figure 8). Interestingly, contrary to their expectations, they discovered that the resulting product was not 2-fluoro-2-deoxy-D-glucose but rather 2-fluoro-2-deoxy-D-allose. The researchers proposed that the acetyl groups at positions C-3 and C-4 were involved in the reaction, giving rise to an intermediate that caused a configuration change in the hydroxyl group at position C-3. Leveraging this method, they successfully synthesized 2-fluoro[^18^F]-2-deoxy-D-allose **50**.

### 2.3. Fluorination of Carbohydrates via Radical Reactions

In addition to nucleophilic and electrophilic reactions, radical reactions have also been employed in the fluorination of sugars. In 2020, Li’s group [64] successfully introduced a fluorine atom at the C-5 position of compound **56** using an Ag(II)-initiated radical reaction (Figure 9). The authors observed that the carbon atom at the C-6 position underwent an elimination reaction. They proposed a potential reaction mechanism as follows: Ag(II) initiates a radical reaction, leading to the transformation of the C-6 hydroxyl group of the sugar into the oxygen radical intermediate **57**. Subsequently, intermediate **58** is formed via β-scission, followed by the fluorination of intermediate **58** to yield the final product **59**. By utilizing this strategy, the group successfully synthesized a range of rare sugars and accomplished the total synthesis of *Clostridium jejuni* tetrasaccharides [65].

Based on the previous work, Li’s group further advanced the method for decarboxylative fluorination of uronic acids, employing a silver radical pathway [66] (Figure 10a). This reaction demonstrates superior substrate scope and compatibility with functional groups compared to dehydroxymethylative fluorination. Notably, it successfully transforms sugars containing benzyl groups and a hexp-(1-4)-hexp moiety. In a recent development, Li’s group established a protocol for the organophotocatalytic synthesis of glycosyl fluorides without the addition of silver species. This method relies on 9-mesityl-10-methyl-acridinium (Mes-Acr^+^)-mediated oxidative decarboxylative fluorination of uronic acids [67] (Figure 10b). The potential of this method is promising in the synthesis of fluorinated nucleosides.

Difluorinated sugars can also be synthesized via the radical addition to glycals. Commonly employed radical initiation conditions include BEt_3_ and metal catalysis, and others. Quirion’s group [68] achieved the successful introduction of 2-(ethoxycarbonyl)difluoromethyl into glycal **64** using BEt_3_ as a radical initiator, resulting in the formation of CF_2_-glycoside **65** (Figure 11).

Radical addition reactions can also be employed in the preparation of trifluoromethyl-modified sugars. In 2015, Ye’s group [69] developed a photocatalytic C-H trifluoromethylation method using glycal **66** as substrates and Umemoto reagent as a trifluoromethyl source. This method exhibits mild reaction conditions, excellent compatibility with various protecting groups (such as acetyl, benzyl, and p-methoxybenzyl), and applicability to glycals featuring different types of sugars (such as glucose, galactose, rhamnose, arabinose, and lactose) (Figure 12).

In 2018, Vincent’s group [70] successfully accomplished the C-H trifluoromethylation of exo-glycal **72** using photoredox and copper catalysis methods. Two different trifluoromethyl sources, A and B, were employed under distinct reaction conditions. The method displayed excellent selectivity in producing Z-configured trifluoromethylation product **73** (Figure 13) while maintaining good compatibility with various protecting groups, such as benzyl, silyl, and acetyl.

In 2021, Ye’s group [71] used electrochemical methods for the first time in the trifluoromethyl modification of glycoconjugates. The authors used glycal **66** as the substrate, a cheap and readily available sodium trifluoromethanesulfonate reagent as the trifluoromethyl source, and added MnBr_2_ as the redox medium, and the reaction was carried out in acetonitrile solution. The method is mild and suitable for a wide range of glycoconjugate substrates protected by different protecting groups (Figure 14).

In 2020, Postigo’s group [72] utilized glycal **83** as a substrate, combined with iodinated perfluoroalkanes as a fluorine source, and employed Ir[dF(CF_3_)PPy]_2_(dtbPy)PF_6_ as a photocatalyst to achieve the synthesis of perfluoroalkyl-modified glycal **84** under blue light conditions (Figure 15).

### 2.4. Fluorination of Carbohydrates via Metal-Catalyzed Coupling Reactions

Pannecoucke’s group [73,74] developed a copper-catalyzed method for the synthesis of difluoroalkylated glycals. The researchers synthesized a series of difluoroalkylation-modified glycals using Cu(PF_6_)(MeCN)_4_ as the catalyst, BrCF_2_CO_2_Et as the difluoroalkylation reagent, and 1,10-phenanthroline as the ligand (Figure 16). This method exhibited good compatibility with various protecting groups (benzyl, acetyl, methyl, etc.) and could be applied to different types of sugars (glucose, galactose, rhamnose, arabinose, etc.). The proposed reaction mechanism suggests that, initially, copper(I) Cu(PF_6_)(MeCN)_4_ undergoes oxidative addition with BrCF_2_CO_2_Et to form compound **93**, resulting in the trivalent copper species. Compound **93** then undergoes an addition reaction with glycal **66**, selectively adding to the electron-rich C-2 position, generating the oxonium intermediate **94**. Subsequently, the base removes a proton to form compound **95**, which is, ultimately, reduced and eliminated to yield the desired product **89**.

The introduction of trifluoromethyl to the glycal was also achieved via metal-catalyzed coupling reactions. Boutureira’s group [75] utilized a 2-iodoglycal **96** as the substrate and CuCF_3_ as the trifluoromethyl source to successfully obtain the trifluoromethyl-modified glycal **97** in a satisfactory yield upon heating at 50 °C (Figure 17). However, this method required the pre-preparation of 2-iodoglycal, making it somewhat tedious and limited in scope. Alternatively, metal-catalyzed coupling reactions were employed to introduce perfluoroalkyl groups into glycal. Boutureira’s group [76] utilized 2-iodoglycal **96** as the substrate and CuC_2_F_5_ as the perfluoroalkyl source, achieving the synthesis of perfluoroalkyl-modified glycal **98** with higher yields under heating at 50 °C.

### 2.5. Fluorination of Carbohydrates via Building Block Strategy

Fluorine-modified sugars can be synthesized using a fluorine-containing building block strategy. In 2007, Mootoo’s group [77] successfully synthesized monofluorinated glycosides using compounds **107** as the initial materials in a four-step tandem reaction, as shown in Figure 18: esterification, Takai reaction, cyclization reaction, and borohydride oxidation reaction. Through variation in the R group, a wide range of fluorinated carbohydrates with different substituents can be obtained.

In addition to the methods mentioned above, fluorocarbohydrates can be obtained by modifying the backbone of the raw material using fluorinated reagents. Figure 19a shows the introduction of difluoroalkyl into the precursor **111** via the Reformatsky reaction, followed by ring closure to achieve difluoro-C-glycosides **113** [78]. Alternatively, the difluoroalkylated product **116** can be obtained by initially introducing two fluorine atoms at the C-2 position of the sugar ring, followed by a one-step ring closure (Figure 19b) [79].

The synthesis of saccharides modified with trifluoromethyl groups using non-sugar compounds as starting materials was initially achieved by the Kobayashi group [80]. They employed the trifluoromethyl-containing compound **117** as the starting material and generated the intermediate product **118** via a Lewis acid-catalyzed aldol condensation reaction. The carbonyl group in **118** was subsequently reduced, yielding the reduced compound. Furthermore, the double bond in the product was oxidized using a dilute potassium permanganate solution to produce the dihydroxy compound **120** (Figure 20a). In a related study, Burger’s group [81] utilized methyl trifluoropyruvate **122** as a source of trifluoromethyl groups. This compound was catalytically coupled with compound **121** using titanium trichloride, leading to the formation of compound **123**. Subsequently, compound **123** underwent a ring-closing reaction under acidic conditions, resulting in the formation of compound **124**. Finally, compound **125** was synthesized in a single vessel via a reductive reaction (Figure 20b).

The synthesis of sugars modified with polyfluorinated groups was achieved using non-sugar compounds that contained polyfluorinated modifications. In 2013, Vincent’s group [82] utilized compound **126**, which featured a CF_2_CF_2_ structure, as the starting material. Through a ring-closing reaction, they successfully obtained the final product **128** (Figure 21).

The above section discusses the chemical synthesis of fluorine-containing glycomimetics. Additionally, fluorinated carbohydrates can be enzymatically produced from chemically fluorinated sugars. Enzymes such as kinases, phosphorylases, and nucleotidyltransferases can mediate this process. A review paper [83] provides a comprehensive overview of the current state of the art in the chemoenzymatic production of fluorinated carbohydrates. Due to space limitations, the detailed content of this review will not be discussed here.

## 3. Impact of Fluorination on the Physical Properties of Carbohydrates

The size of fluorine atoms is comparable to that of oxygen atoms, but they have a higher ionization potential. This higher ionization potential makes fluorine atoms less capable of accepting hydrogen bonds compared to oxygen atoms. As a result, the C-F group is more hydrophobic than the alcohol group. Introducing fluorine atoms into molecules can aid in enhancing biomolecular recognition via the process of hydrophobic desolvation. Carbohydrates have a strong hydrogen-bonding network with solvent molecules, and their low binding affinity can be attributed to the high energy required for desolvation. Deoxyfluorination, the process of removing hydroxyl groups and replacing them with fluorine atoms in carbohydrates, can improve the binding affinity of these molecules towards target units. The high electronegativity of fluorine limits the polarizability of the C-F bond, reduces the solvation of the ligand, increases hydrophobicity, decreases the penalty of desolvation, and enhances binding affinity. Additionally, polar C-F bonds maintain necessary electrostatic interactions between ligands and binding sites such as cations and dipoles.

The C-F bond (1.35 Å) is longer than the C-H bond (1.09 Å) and shorter than the C-OH bond (1.43 Å). CF_2_ is considered a more favorable option compared to CHF for replacing CHOH, as it has lower steric and desolvation costs. Monodeoxyfluorinated sugar mimics showed improved affinity for various targets after modification, and a significantly high affinity was observed when the compounds were modified with polydeoxyfluoride. It has been observed that hexose analogs containing multiple fluorine substitutes exhibit stronger binding compared to normal substrates. For example, 2-deoxy-2,2-difluoro-D-glucose showed preferential binding to yeast hexokinase, and 2-deoxy-2-fluoro-α-D-glucose (**129**) displayed enhanced inhibition of glycogen phosphorylase [12] (Figure 22a). Another example is hexa-fluorinated-glucose derivatives such as 1-hydroxy-5-hydroxymethyl-2,2,3,3,4,4-hexafluorane (**132**), which exhibited approximately ten times higher permeability across red blood cell membranes compared to 3-deoxy-3-fluoro-D-glucose (**133**), indicating enhanced affinity for the transporter [9]. These results indicate the importance of the polar hydrophobicity of fluoro-sugars.

### 3.1. The Impact on Lipophilicity

The lipophilicity of a compound is determined using its partition coefficient (P), which is measured between 1-octanol and water and expressed as logP. Deoxyfluorination induces increased hydrophobicity, resulting in a significant rise in logP. The magnitude of this change is dependent on the position and number of fluorine atoms integrated into the molecule. The Linclau group employed a ^19^F NMR-based approach to determine logP values for various deoxyfluorinated carbohydrate derivatives (Figure 23) [84,85]. Generally, each deoxyfluorination leads to a logP increase of approximately 1 unit. The site of fluorination (e.g., 6-deoxy-6-fluoro-glucose, logP = −2.36) and the stereochemistry of the carbohydrate (e.g., 2-deoxy-2-fluoro-galactose, logP = −2.37) also influence the logP values. The lipophilicity of 2,3,4-trideoxy-2,3,4-trifluoro-monosaccharide derivatives can vary considerably [31], and the all-cis derivative exhibits the lowest hydrophobicity, likely due to its large molecular dipoles.

St-Gelais and colleagues recently conducted a comprehensive investigation into the lipophilicity of monofluorinated, difluorinated, and trifluorinated glucopyranose analogs, with modifications at positions C-2, C-3, C-4, and C-6. This study complements the previous work by Linclau and colleagues, who also explored the lipophilicity of fluorinated carbohydrates. In general, analogs bearing fluorine atoms at C-6 exhibit the highest hydrophilicity, while compounds with vicinal polyfluorinated motifs display the greatest lipophilicity. This observation can potentially be attributed to the considerable reduction in the polar surface area of the molecules [86]. The calculated solvation-free energy demonstrates a good correlation with the experimentally determined log P values. Although the lipophilicity conferred by polyfluorination on sugars has the potential to improve affinity and membrane permeability, an excessive degree of lipophilicity can compromise the efficacy and safety of drugs, leading to increased attrition rates [87,88]. Therefore, striking an appropriate balance is crucial.

### 3.2. The Impact on Conformation

The small van der Waals radius and high electronegativity of the fluorine atom result in the formation of a C–F bond with a strong dipole moment (μ C–F = 1.41 D) and a low-lying C–F σ* orbital available for hyperconjugation. These factors have various effects on the conformation of the molecules. The preference for certain conformations can be exploited to stabilize a desired binding conformation and improve affinity. However, it can also inadvertently favor an undesired conformation, which negatively affects binding [89,90].

#### 3.2.1. The Impact on the Chair Conformation of Pyranose

The conformation of fluorinated carbohydrate derivatives has been extensively studied using X-ray diffraction and NMR techniques [31]. In general, the introduction of an ectopic F atom affects the ^4^C_1_/^1^C_4_ equilibrium, and it has been observed that fluorination does not result in a significant degree of distortion in the pyranose conformation, even when the compounds are multifluorinated (e.g., **139**) (Figure 24). The review article [31] published by Linucau et al. provides an excellent summary of the different ways in which fluoridation affects carbohydrate structure and conformation. Analysis of the crystal structures of free trideoxytrifluorinated sugars such as glucose **140** [91,92] and the corresponding altrose and galactose molecules (all β-anomers) reveals that these molecules present ^4^C_1_-conformations. However, slight distortions in the structures are possible when 1,3-coaxial C-F bonds are present in the molecules, as shown for **141** [93,94]. Hexafluoropyranose **142** presents the ^4^C_1_-conformation, as the repulsive forces between the two F atoms cause them to move away from each other. Conformational analysis of α-and β-difluoro-glucose, galactose, and mannose derivatives reveals that the CF_2_ unit in the ring does not significantly influence the properties of the ^4^C_1_ chair. The corresponding alkyl bioisostere of a CF_3_ group has been studied, and it has been reported that based on the size and shape of the group, ethyl or an isopropyl group is the closest match [95,96]. In pentopyranosyl fluorides, the strong anomeric effect involving the anomeric C–F bond promotes chair inversion. This phenomenon has also been observed in the case of the triacetylated β-D-xylopyranosyl fluoride **143**, where ring inversion results in the generation of a structure with three axial ester groups.

#### 3.2.2. The Impact on the Conformation of the Glycosidic Bond

In O-glycosides **144**, stereoelectronic stabilization caused by an overlap between the lone pairs of electrons on the anomeric oxygen and the σ*_C1–O1_ orbital (the exo-anomeric effect) and steric repulsion result in the generation of a specific exo conformation of the glycosidic bond (Figure 25a). Solitary pairs on the outer ring atoms are absent in CH_2_-glycosides, and the polarity of the C1–C2 and C1–C5a units are comparable. The conformation distribution is influenced by the minimum spatial repulsion force. However, the formation of CF_2_-glycosides introduces the possibility of the generation of the fluoro-gauche effect. This stereoelectronic effect (Figure 25b) is based on the hyperconjugation of the C–F bond to the C1–H1 and C1–C2 bonds and is closely related to the ectopic effect that influences the conformation of fluorinated molecules [97]. Figure 25b presents the schematic representation of the two conformations, **145** and **146**, that allow the generation of the most stable hyperconjugated states. The evaluations of the ectopic effects exerted by CHF- and CF_2_-glycosides indicate the predominance of non-natural conformations. The CF_2_-glycosidic linkage has been incorporated into several different glycomimetics structures. The CF_2_-linked α-(2,3)-sialylgalactose compound has been used as a non-hydrolyzable mimic of sialylated gangliosides (such as GM3 and GM4). A CF_2_-linked GM4 derivative exhibited inhibitory activity against NEU2 (IC_50_ = 754 mm) and NEU4 (IC_50_ = 930 mm). It was observed that this compound successfully inhibited the proliferation of human lymphocytes [98].

In carbasugars, C1–C2 and C1–C5a have the same polarity; therefore, no preferential conformation is observed. Leclerc et al. proposed a CF_2_ analog of carbasugars in which the CF_2_ group replaced the endocyclic oxygen atom to restore the exo conformation attributable to a polarization of the C_1_–CF_2_ bond [99] (Figure 26). Carbasugars lack a hemiacetal structure, making these derivatives significantly less susceptible to hydrolysis than derivatives with an exocyclic CF_2_ group replaced. Studies have shown that the replacement of the endocyclic oxygen atom with a CF_2_ group in maltose restored stereoelectronic stabilization caused by the lone pairs of electrons on the anomeric oxygen and the σ*_C1–O1_ orbital. The replacement resulted in the preferential generation of the exo conformation [100]. The electron-absorbing CF_2_ group replaces the endocyclic CH_2_ unit and reduces the electron density on the endocyclic-O atom, thus reducing the degree of the endo-anomeric effect. This suggests that electron-absorbing substituents can reduce the strength of the spatial torsion interaction and help maintain a natural conformational equilibrium. Fluorinated carbasugars have the added advantage of increasing the overall lipophilicity, but this property must be carefully evaluated as they can also influence the acidity of neighboring functional groups and potentially generate novel O–H···F intramolecular interactions or 1,3-diaxial steric clashes.

The conformational mobility of the iduronic acid (L-IdoA) moiety in heparin oligosaccharides, both in free and bound states, has been studied. Distinct conformations of the L-IdoA ring are recognized depending on the protein receptor [101]. For instance, antithrombin III (AT-III) recognizes the skew boat conformer [102], while the L-IdoA rings of a heparin hexasaccharide maintain the chair–skew boat flexibility when bound to the fibroblast growth factor-1 (FGF-1) unit [103]. Fluorination can significantly affect the biological activity of IdoA-based glycomimetics, and this can be attributed to the conformational plasticity of L-IdoA that mediates receptor binding. Gem-difluorocarbasugar analogs based on L-Ido and Glc (control) have been studied using a combination of NMR spectroscopy and computational techniques. The results demonstrate that the fluorocarbasugar L-IdoA analog closely mimics the conformational plasticity of native Ido, whereas the non-fluorinated carbasugar was found to be incapable of retaining this conformational flexibility [104].

#### 3.2.3. The Impact on the Conformation of the Exocyclic Hydroxymethyl Group

The C4 configuration plays a significant role in determining the exocyclic conformation of the C5–C6 units, and conformations characterized via a regimented arrangement of C–O bonds are usually unfavorable, and this is expected to be similar in the case of deoxyfluorination. The preferred conformation of the heterocyclic fluoromethyl groups in 6-deoxy-6-fluoroglucose is shown in Figure 27. The gt conformation is the most abundant for the galactose derivative **149**, and the tg conformation is present in a small amount in this system. The *gt* conformation is proposed to be the dominant species in solution for the 4,6-dideoxy-4,6-difluorinated GalNAc derivative **150** [105], while the *gg* conformation is the predominant configuration of the 4,6-difluorinated glucose derivative **151** [106]. Interestingly, the crystal structure of the 2,3,4,6-tetrafluorinated galactose derivative **152** presents the *gg* conformation. Therefore, it was proposed that the *gt* conformation was present in the solution phase [90].

### 3.3. The Impact on the Stability of the Glycosidic Bond

The high electronegativity and inductive ability of fluorine make it difficult for fluorinated carbohydrate derivatives to form oxocarbenium moieties [106]. This was verified in hydrolysis studies of glycosyl phosphates (where the observed order of phosphate cleavage was native, **157** > 6-fluoro, **156** > 3-fluoro, **155** > 4-fluoro, **154** > 2-fluoro, **153** (Figure 28) and dinitrophenylglycosides. However, the order was reversed (2-deoxy, **160** > 4-deoxy, **159** > 3-deoxy, **158** > native, **157**) in a series of deoxygenated alkyl and aryl glucoside mimetics [107,108,109]. As both enzymatic and acid-catalyzed glycosidic cleavage typically proceeds through an oxocarbenium-like intermediate, all positions of the sugar molecule were fluorinated to improve hydrolytic stability and/or prevent enzymatic cleavage. Fluorinated glycomimetics are often used as excellent inhibitors of enzymes, and this is important for improving drug bioavailability.

The destabilizing effect on the process of formation of the oxocarbenium transition state has been widely exploited for designing inhibitors for GTs [110,111,112,113,114] and glycosidases. 2,4-dinitrophenyl-2-deoxy-2-fluoro-β-glucopyranoside **161** (Figure 29a) was the first example of a fluorinated glycoside inhibitor reported by the Withers laboratory [115]. A number of differently configured fluorinated glycosides have since been synthesized, which are hydrolyzed via the target enzyme in a manner similar to that of a natural substrate. The presence of the electron-withdrawing fluorine substituent at C-2 destabilizes both the glycosylation and the deglycosylation transition states. As a result, the rates with which the enzyme–substrate intermediate is formed as well as hydrolyzed are decreased. To compensate for the overall deactivation of the inhibitor, an activated anomeric leaving group (such as dinitro or trinitrophenate or fluoride) is usually introduced to accelerate the glycosylation step, resulting in the accumulation of covalent glycosyl–enzyme intermediates, thereby inhibiting the enzyme. These inhibitors have been shown to inhibit various types of retaining glycosidases [116,117,118,119]. The success of 2-fluoro-2-deoxyglucose as the most utilized radiotracer used for positron emission tomography (PET) can be attributed to its resistance to hydrolysis. This helps the molecule avoid metabolic degradation and enables tissue accumulation. The effects of fluorination on anomeric reactivity are presented for fluorinated maltoses **163** [120] and **164** [121] (Figure 29b). The GlgE1 maltosyl transferase from Streptomyces coelicor was studied to investigate these molecules as mechanism-based enzyme inhibitors. It was observed that compound **163** formed covalent intermediates by reacting with relevant nucleophilic Asp residues [120]. The enzyme was able to effect hydrolysis. This could only be prevented under conditions of an E423A point mutation. However, trifluorinated maltose **164** exhibited increased deactivation properties resulting from C2-difluorination, which made the compound unreactive to the enzymes [121]. It is interesting that **164** can crystallize with GlgE1 in the complex.

Phlorizin is a natural product isolated from apple tree bark with inhibitory activity against the sodium-glucose linked transporters (SGLT1 and SGLT2) (Figure 30a) and has been studied as a therapy for type II diabetes mellitus. Phlorizin has been associated with rapid hydrolytic degradation, and there is much interest in achieving selective SGLT2 inhibition. 2-Deoxy-2-fluoro derivatives of Phlorizin analogs **165** were evaluated in efforts to improve its metabolic stability and improved resistance to hydrolysis and were also observed to improve selectivity for SGLT2 (albeit with lower affinity) [122], which made it a better substrate for metabolic enzymes. In another study, fluorinated sialic acid analogue (P-3F_ax_-Neu5Ac) **166** (Figure 30b) was used to inhibit sialyltransferase activity since its inherent electron-withdrawing capability significantly increases the electrophilicity of the C-2 atom and destabilizes the oxocarbenium transition state. This destabilization hinders its formation, significantly slowing the enzymatic rate and effectively reducing tumor growth in vivo [123,124].

## 4. Application

### 4.1. Fluorine-Containing Sugars as Glycosyltransferase Inhibitors

#### 4.1.1. Fluorosugars as in Cellulose Glycosyltransferase Inhibitors

Glycosyltransferase (GT) participates in the glycosylation of glycosylated receptor substrates and helps in assembling oligosaccharides. Nucleotide–sugar units are used as the activated sugar donor substrates during the process. The inhibition of GT has been a focus of research due to its biological relevance [125,126,127,128,129].

The fluorinated sialic acid analog, which acts as a transition-state inhibitor for sialic transferase, is a GT inhibitor [110]. Although it cannot cross cell membranes due to its high hydrophilicity, some of the precursors (**167**) can be transported across the membranes into the cytoplasm and enzymatically converted to fluorinated sugar-nucleotides, resulting in the in situ production of GT inhibitors (Figure 31a). For example, peracetylated 3-fluorosialic acid **168** (Figure 31b) [130] acts as a specific inhibitor for sialic transferase, and it can penetrate cell membranes. Studies have shown that this inhibitor can hinder the process of tumor growth and metastasis in vitro and in vivo [123,124,131]. A second-generation pre-fluorination inhibitor **169** with a carbamate group at C-5 was synthesized. This compound exhibited effective and increased SialT inhibitory activity, and this could be attributed to the efficient intracellular conversion of the molecules to their active nucleotide sugar forms [132].

These results indicate that **168** is an effective sugar-like agent, and the peracetylation form of the compound does not affect the process of sialylation. It can be inferred that the fluorine atoms in the inhibitor structure impart the inhibitory effect to the molecule. The results suggest that fluorinated sialic acid analogs inhibit the process of sialacidification and are potential tools for anticancer therapy [124].

Linclau and coworkers thoroughly investigated the process of formation of 2-deoxy-2-fluoro-fucose (GDP **170**, Figure 32) in cellulose [110,130,133]. The results revealed that the accumulation of nucleotide sugar **170** not only inhibited the activity of FucTs but also shut down the biosynthesis of GDP-fucose (natural FucT donor) [130].

Recently, efforts have been made to produce non-fucosylated recombinant antibodies in CHO cells by blocking fucosylation. This was achieved by inhibiting the de novo biosynthesis of GDP fucose via the hydrolysis of petrosterol, which resulted in the transformation of peracetylated fucostatin to GDP fucostatin [134]. Non-fucosylated therapeutic antibodies are reported to exhibit better antibody-dependent cytotoxicity and improved in vivo efficacy. This blocking strategy was also used to generate low-fucosylated monoclonal antibodies from murine hybridoma cells [135]. Monofluoride and difluoride galactoside **171** (Figure 33) inhibited the biosynthesis of UDP-galactose in epithelial cells and fibroblasts, resulting in the production of metabolically induced galactose phenotypes.

Recently, a monofluorinated analog of 2,4-diacetamide-2,4,6-trideoxygalactose, which is typically found only in bacteria, was found to inhibit the ability of H. pylori to synthesize glycoproteins significantly. This resulted in a reduction in its growth and motility and a decrease in the extent of biofilm formation [136]. The proposed mode of action involves the in situ generation of glycosyltransferase inhibitors.

#### 4.1.2. Fluorine-Containing Sugars as Glycosidase Inhibitors

Glycosidase is a family of hundreds of enzymes that hydrolyze glycosidic bonds, forming monosaccharides in living organisms. The dysfunction of glycosidase is directly related to the etiology of diseases such as diabetes. Therefore, glycosidase inhibitors are potential drugs for treating these diseases [137]. Enzyme-catalyzed hydrolysis with glycosidase retention occurs via the production of glycosylase intermediates with oxacarbenium ion-like transition states. When the highly electronegative fluorine atom replaces the hydroxyl group in the molecules (especially the group at the C-2 site) of the carbohydrate ring, the fluorine atom inhibits the formation and hydrolysis of intermediates. This can be attributed to the inducing effects exerted and the instability of the oxacarbenium ion-like transition states [138]. This results in a significantly reduced rate of glycosylation and deglycosylation. In vivo studies have demonstrated the inhibitory effect of 2-deoxy-2-fluoro-β-glucopyranoside and flumannoside on β-glucosidase and β-mannosidase [139].

2-Fluoro-deoxysugars, particularly fluoro-sialosides represented as 2,3-difluorosialoside (DFSA, trypanosoma Cruzi trans-sialase (TcTs) inhibitors) were used to study the mechanism of action of glycosylated hydrolases [140,141]. In 2013, Wong and co-workers developed DFSA **172** (Figure 33a), which contained acetylene as functional groups at five sites. It was found to be an irreversible inhibitor of viral, bacterial, and human sialidase [142]. Withers et al. studied a group of DFSAs (**173**; Figure 33a) as compounds with antiviral activity. They also tested the compounds for anti-influenza activity. DFSAs form a potent class of inhibitors that act through the instantaneous formation of covalent intermediates. As an enzyme, they exhibit high efficiency, and the degree of antiviral activity of these compounds is comparable to or higher than that of Zanamivir [143,144]. Notably, tyrosine residues in DFSAs **172**–**174** played a catalytic role as a nucleophile. Since the discovery of nojirimycin **175** and its derivatives **176** (Figure 33b) [145,146], nitrogen-containing carbohydrate analogs have played important roles as inhibitors of glycoside hydrolase. Given the activity of iminosugars, many research groups have been trying to synthesize fluorinated analogs of iminosugars like **177** to achieve stronger and highly selective inhibition properties, which were summarized in previously conducted reviews [147,148,149,150].

### 4.2. Synthetic Vaccines and Therapeutic Antibodies

Synthetic vaccines containing natural sugar antigens face the problem of enzymatic degradation in vivo. However, this issue can be overcome using fluoridation. Fluorine has unique properties that can enhance ligand binding affinity and improve immunogenicity. The similar size of fluorine and hydrogen allows for the elicitation of cross-reactive antibodies, while the super-hydrophobicity of fluorocarbons and the unique induction effect and “polar hydrophobicity” of fluorine are often utilized to enhance the binding affinity of the molecules. The process of antigen fluorination has become an effective means of improving the affinity of TCR without significantly interfering with antigen composition or structure.

Epithelial tumor cells are often present as truncated and/or hyper-sialylated glycan in glycoproteins. These glycan structures are often the target of anticancer vaccines. However, these exhibit poor metabolic stability. Hoffmann-Roder and Kunz’s groups reported that fluorinated Thomsen–Friedenreich (TF) conjugates, such as **178** and **179** (Figure 34), generated a strong immune response in mice, and their antibodies were shown to bind strongly to epithelial tumor cells (MCF-7) [25,26,27,151,152,153]. Subsequently, it was found that immunization against a wider range of TF-MUC analogs fluorinated at the 6- and 2- positions produced antisera that exhibited little difference in their binding ability to bind to fluorinated and nonfluorinated antigens.

Our research group conducted a systematic and thorough investigation into the structural modification of STn antigens [154,155,156]. Three N-acetyl-fluoro-modified derivatives (Figure 35) were conjugated with keyhole limpet hemocyanin (KLH), and these generated a high titer immune response. The degree of response was 3–5 times higher than the degree of response generated via the natural STn–KLH system. These N-acetyl-fluoro-modified STn derivatives were site-modified onto the MUC1 glycopeptide moiety and conjugated with KLH. The results from immunological experiments revealed that the structurally modified glycoantigen could potentially improve the immunogenicity of the MUC1 glycopeptides [157]. Other TACAs, such as Tn [158], TF [159], GM3 [160], and KH-1 [161], were also fluorinated and conjugated with protein carriers to develop vaccines, which exhibited strong immune cross-reaction.

Globo H is a tumor-associated carbohydrate antigen (TACA), and clinical trials have been conducted to evaluate vaccines based on Globo H for cancers, including breast, ovarian, and prostate cancer [162]. This vaccine typically induces higher IgM titers (compared to IgG titers). A deoxyfluoro derivative of Globo H was developed that elicited IgG antibodies with titers comparable to those elicited by native Globo H to address this issue. This derivative also recognized Globo H, its related epitopes (stage-specific embryonic antigens 3 and 4), and Globo H-expressing tumor cells [162], leading to complement-dependent cell cytotoxicity.

The process of fluoridation is also being utilized to develop HIV vaccines. Efforts have been made to cluster sugar antigens to enhance their immunogenicity and affinity for 2G12. For example, Danishefsky [163] linked oligomannosaccharide Man_9_(GlcNAc)_2_ to OMPC via a cyclic peptide skeleton, which resulted in the formation of a sugar complex that could strongly react with 2G12 antibodies [164]. However, this complex did not induce the production of the corresponding bnAbs units in guinea pigs and rhesus monkeys [165]. Studies have shown that inoculating different mixtures of immunogens with the same epitope can improve the antibody’s affinity toward antigen epitopes. Based on this, our research group developed a highly convergent and effective synthesis strategy for Man5 and its monofluoro-modified, trifluoro-modified, and S-linked analogs, which could be combined with CRM197 to develop an anti-HIV vaccine candidate [166]. The results revealed that the modified analogs elicited strong antibody responses. However, the induced antibodies were unable to bind to the native gp120 antigen. This could be potentially attributed to immunotolerance mechanisms that suppress the generation of immune response for Man5-related structures and the conformation of sugar epitopes on synthetic glycoconjugates. These conformations differed from the conformations of natural sugar epitopes on gp120. The results provide significant insights into the process of development of HIV vaccines.

## 5. Conclusions

Carbohydrates are widely found in nature and have significant involvement in physiological processes. Therefore, the development of methods to control these processes is of great interest for therapeutic applications. However, progress in the field of carbohydrate-based drugs has been limited. This can be attributed to their high polarity, resulting in low binding affinities, as well as their poor pharmacokinetic properties, such as short residence time and rapid renal excretion. In recent years, fluorine has been introduced into more than one-third of newly developed small-molecule drugs. This has provided valuable insights into the design of glycomimetics as a potential solution to address the challenges associated with carbohydrate-based drugs. Despite there being growing evidence of the environmental accumulation of certain per- and polyfluorinated alkyl substance materials and further concerns for their bioaccumulation and potential toxicological effects [167], fluorine modification continues to be irreplaceably important in new drug research and development.

This review focuses on the synthesis of fluoro-sugars and explores the effects of fluorination on their properties and biological activities. Extensive research has been conducted on various methods for synthesizing fluorinated glycomimetics, including nucleophilic substitution, electrophilic addition reactions, and radical fluorination. However, there remains a need for the development of milder, stereoselective, and regioselective fluorine substitution methods. Substituting fluorine for the hydroxyl group can maintain certain electrostatic interactions while reducing overall polarity through increased hydrophobicity. This modification has demonstrated significant utility in the development of glycosidase inhibitors, synthetic vaccines, and therapeutic antibodies. It enhances residence time and binding affinity, thereby improving the efficacy of these compounds.

Due to the extensive body of research on glycomimetics and the acknowledged significance of carbohydrates in transportation and delivery, there exists considerable potential to investigate the possible biological activities of fluoro-sugars, provided that appropriate fluorinated modifications are undertaken. Furthermore, given the intensive research on glycan synthesis and biological functions, the development of fluorinated glycomimetics is anticipated to enhance the application of carbohydrates in diagnostic, therapeutic, and vaccine research.

## Figures and Tables

**Figure 1 molecules-28-06641-f001:**
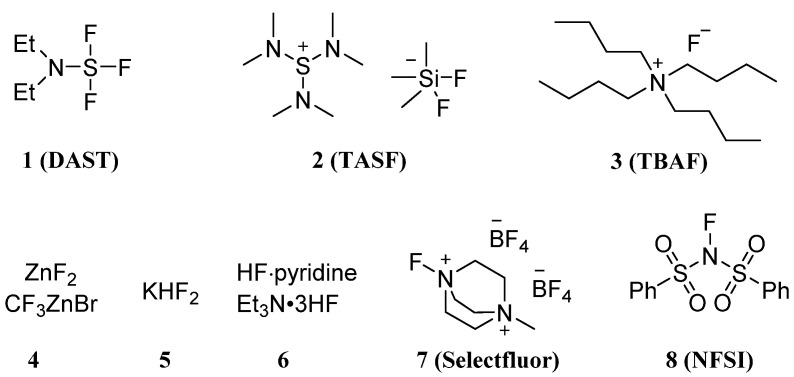
Typical fluorinating reagents: **1**, DAST (diethylaminosulfur trifluoride); **2**, TASF ((tris(dimethylamino)sulfonium difluorotrimethylsilicate), **3**, TBAF (tertbutylammonium fluoride); **4,** Metal fluorides; **5**, KHF_2_; **6**, anhydrous hydrogen fluoride system; **7**, SelectFluor, (1-chloromethyl-4-fluoro-1,4-diazoniabicyclo[2.2.2]octane bis(tetrafluoroborate) and **8**, NFSI (N-fluorobenzenesulfonimide).

**Figure 2 molecules-28-06641-f002:**
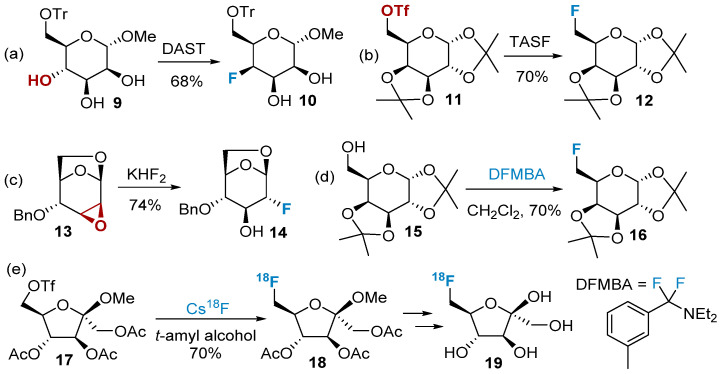
Nucleophilic fluorination of carbohydrates using different fluorinating reagents: (**a**) DAST as fluorinated reagents, (**b**) TASF as fluorinated reagents, (**c**) KHF_2_ as fluorinated reagents, (**d**) DFMBA as fluorinated reagents, (**e**) Cs^18^F as fluorinated reagents.

**Figure 3 molecules-28-06641-f003:**
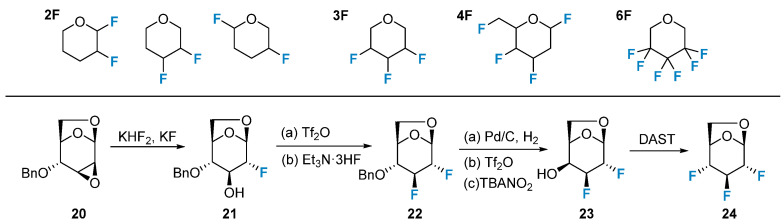
Polyfluorination of carbohydrates via iterative nucleophilic reactions and the synthesis of 2,3,4-trifluoro-glucopyranose.

**Figure 4 molecules-28-06641-f004:**
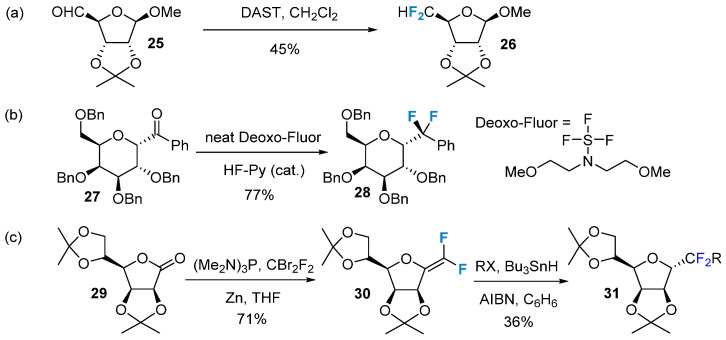
Difluorination of carbohydrates via nucleophilic reactions. (**a**) nucleophilic difluoroalkylation via DAST, (**b**) nucleophilic difluoroalkylation via Deoxo-Fluor, (**c**) nucleophilic difluoroalkylation via CBr_2_F_2_.

**Figure 5 molecules-28-06641-f005:**
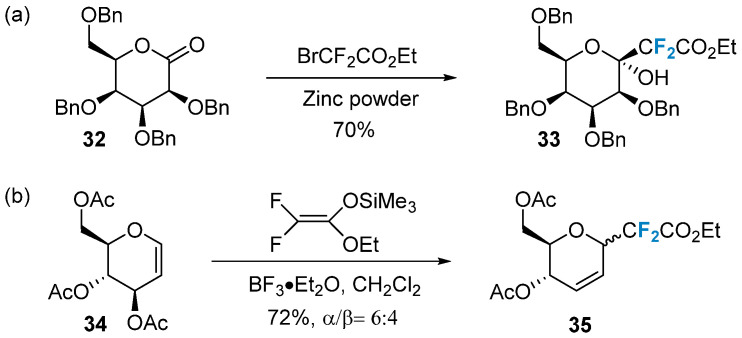
Difluoroalkylation of sugars via nucleophilic reactions. (**a**), difluoroalkylation via BrCF_2_CO_2_Et, (**b**) difluoroalkylation via CF_2_CO_2_Et.

**Figure 6 molecules-28-06641-f006:**
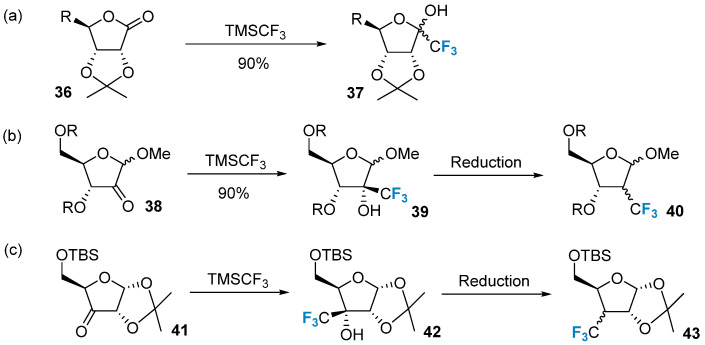
Trifluoromethylation of sugars via nucleophilic reactions. (**a**) trifluoromethylation via the nucleophilic addition of TMSCF_3_ to ketone on C-1, (**b**) trifluoromethylation of ketone on C-2, (**c**) trifluoromethylation of ketone on C-3.

**Figure 7 molecules-28-06641-f007:**
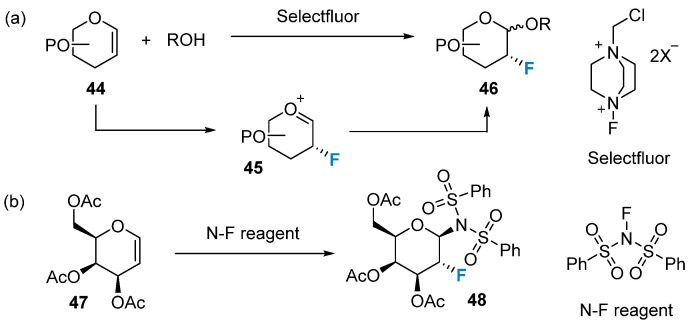
Monofluorination of glycals using N-F reagents. (**a**) monofluorination via Selectfluor, (**b**) monofluorination via N-fluorobenzenesulfonamide.

**Figure 8 molecules-28-06641-f008:**
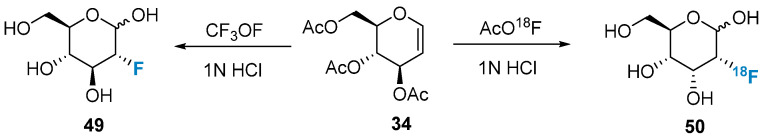
Monofluorination of glycals using O-F reagents.

**Figure 9 molecules-28-06641-f009:**
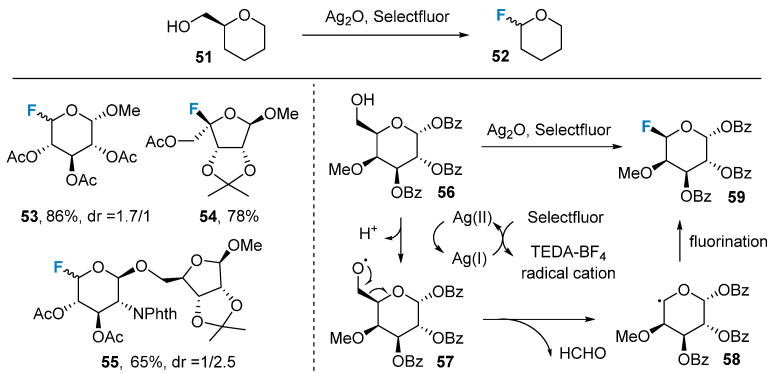
Dehydroxymethylative fluorination of sugars via Ag(II)-initiated radical reaction.

**Figure 10 molecules-28-06641-f010:**
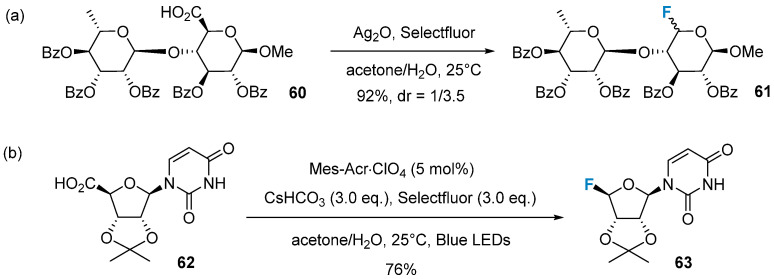
Decarboxylative fluorination of uronic acids via radical reaction. (**a**) decarboxylative fluorination via silver radical pathway, (**b**) Mes-Acr^+^-mediated oxidative decarboxylative fluorination.

**Figure 11 molecules-28-06641-f011:**
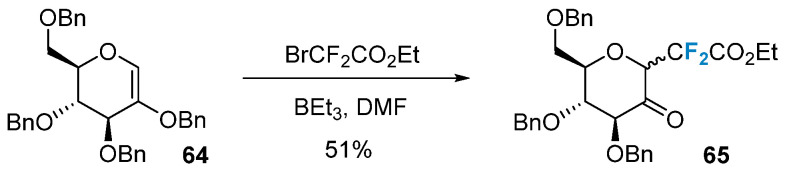
Difluoromethylation of glycals via radical reaction.

**Figure 12 molecules-28-06641-f012:**
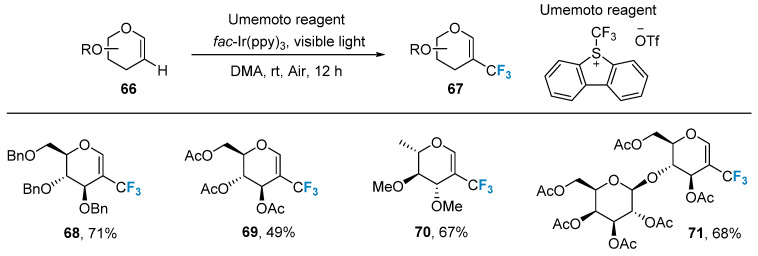
Trifluoromethylation of glycals using Umemoto reagent.

**Figure 13 molecules-28-06641-f013:**
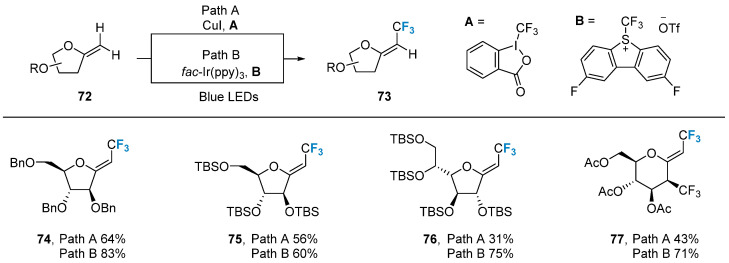
Trifluoromethylation of glycals via photoredox and copper catalysis.

**Figure 14 molecules-28-06641-f014:**
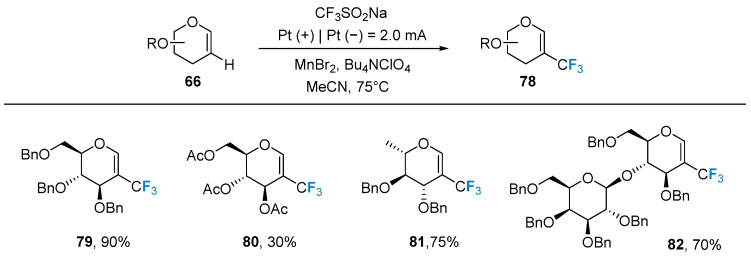
Trifluoromethylation of glycals via electrochemical method.

**Figure 15 molecules-28-06641-f015:**
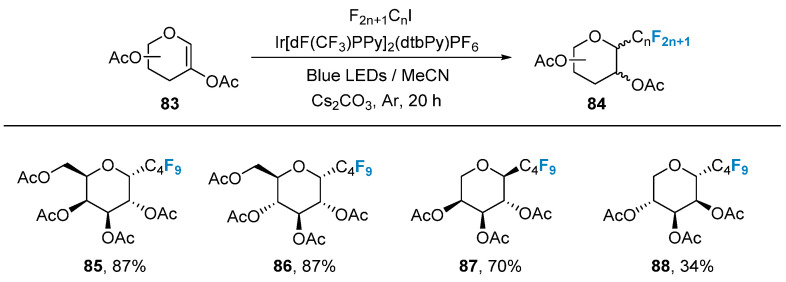
Photocatalyzed reductive fluoroalkylation of 2-acetoxyglycals.

**Figure 16 molecules-28-06641-f016:**
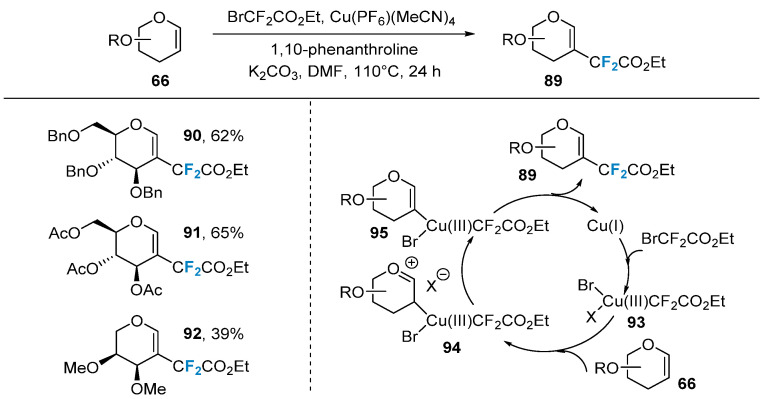
Copper-catalyzed β-difluoroacetylation of glycals via direct C–H functionalization.

**Figure 17 molecules-28-06641-f017:**
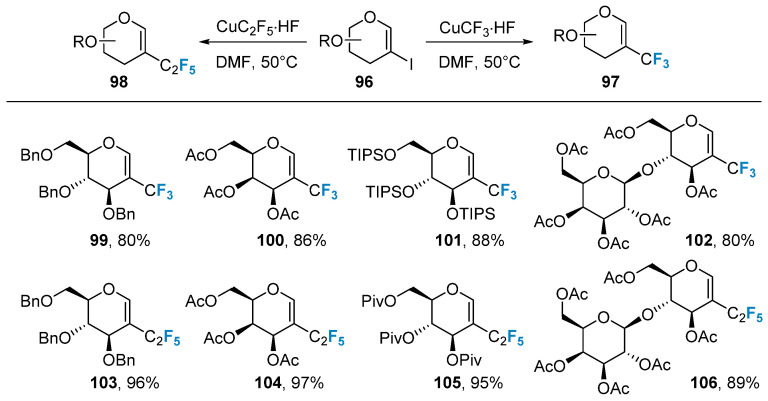
Copper-catalyzed trifluoromethylation of 2-iodoglycals.

**Figure 18 molecules-28-06641-f018:**
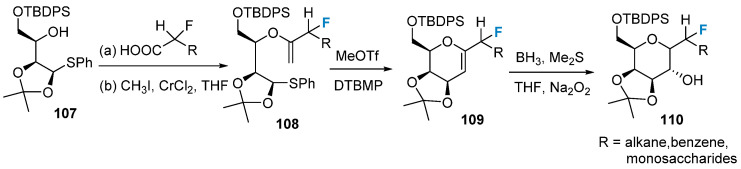
Monofluorination of sugars via building block strategy.

**Figure 19 molecules-28-06641-f019:**
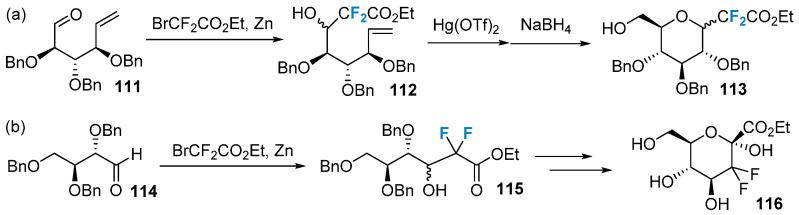
Difluorination of sugars via building block strategy: (**a**) synthesis of difluoro-C-glycosides **113** via building block strategy, (**b**) synthesis of difluoroalkylated product **116** via building block strategy.

**Figure 20 molecules-28-06641-f020:**
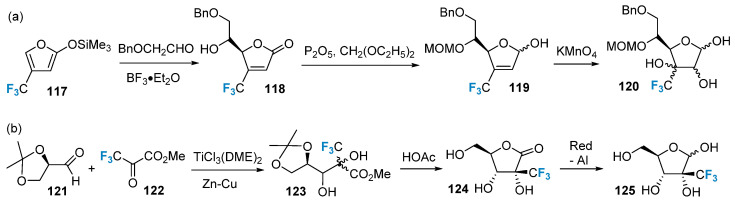
Trifluoromethylation of sugars via building block strategy: (**a**) synthesis of trifluoromethylated product **120** via building block strategy, (**b**) synthesis of trifluoromethylated product **125** via building block strategy.

**Figure 21 molecules-28-06641-f021:**
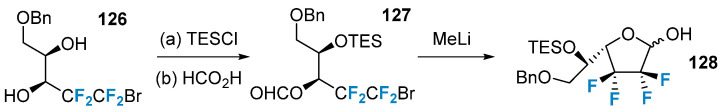
Polyfluorination of sugars via building block strategy.

**Figure 22 molecules-28-06641-f022:**
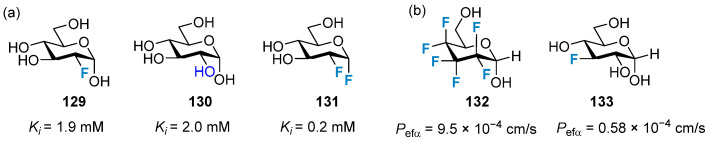
(**a**) Inhibitors of glycogen phosphorylase exhibited an enhanced binding affinity when modified with difluorinated derivatives. K_i_ = inhibition constant values. (**b**) The polyfluorinated analog exhibits enhanced diffusion characteristics. P_ef_ = efflux permeabilities.

**Figure 23 molecules-28-06641-f023:**
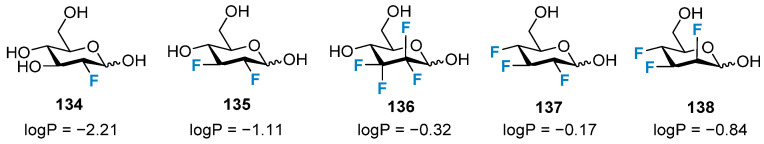
Lipophilicities of fluorinated sugars.

**Figure 24 molecules-28-06641-f024:**
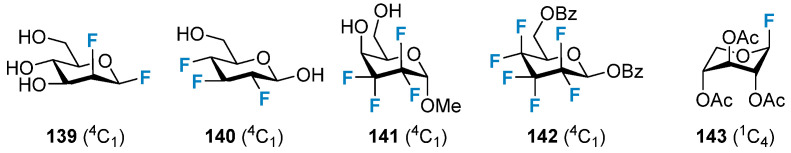
Ring conformation of fluorinated carbohydrate derivatives.

**Figure 25 molecules-28-06641-f025:**
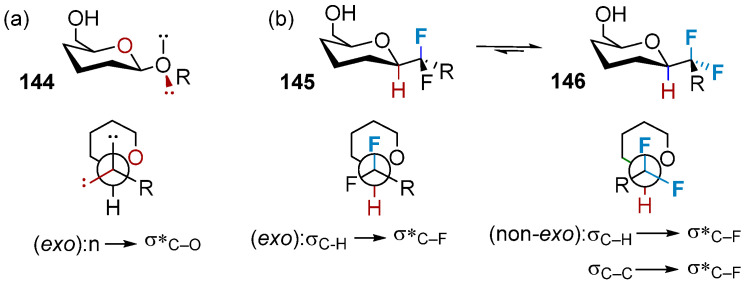
The anomeric and fluorine gauche effects determining anomeric conformation. (**a**) exo conformation of the glycosidic bond in O-glycosides, (**b**) two conformations allow for the most stabilising hyperconjugation situations in CF_2_-glycosides.

**Figure 26 molecules-28-06641-f026:**
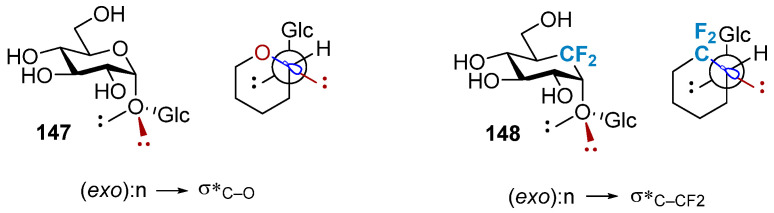
Schematic representation of the lone pair–s* interactions.

**Figure 27 molecules-28-06641-f027:**
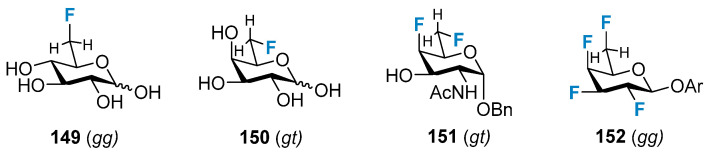
Preferred conformations of the exocyclic fluoromethyl group.

**Figure 28 molecules-28-06641-f028:**
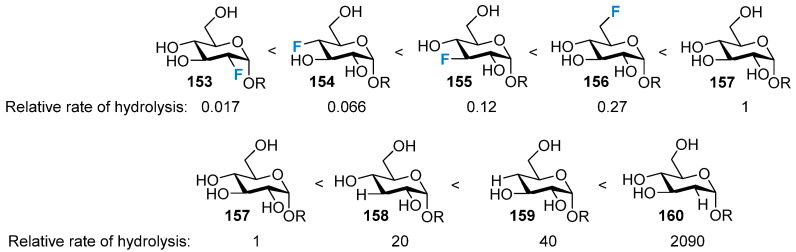
The relative rates of acid-catalyzed hydrolysis of deoxy- and deoxyfluoro-α-glucopyranoside derivatives.

**Figure 29 molecules-28-06641-f029:**
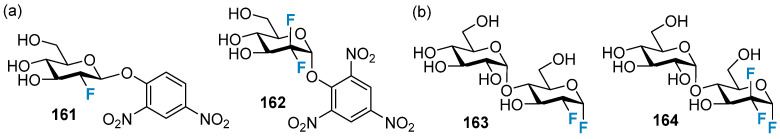
Fluorinated maltose derivatives as mechanism-based inhibitors. (**a**), 2,4-dinitrophenyl-2-deoxy-2-fluoro-glucopyranoside as fluorinated glycoside inhibitor, (**b**) 2-fluoro-2-deoxyglucose as radiotracer for PET.

**Figure 30 molecules-28-06641-f030:**
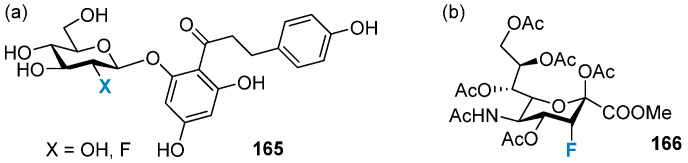
(**a**) Phlorizin and its 2-deoxyfluoro derivative as SGLT inhibitors; (**b**) P-3F_ax_-Neu5Ac inhibiting sialyltransferase activity.

**Figure 31 molecules-28-06641-f031:**
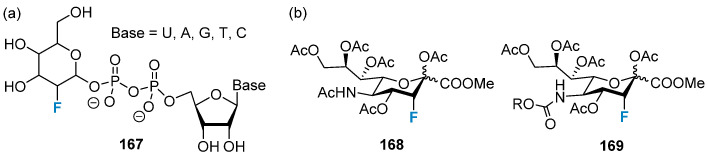
(**a**) Fluorinated nucleotide–sugars as enzyme inhibitors or inactivators. (**b**) 3-Fluoro-sialic acid and its second-generation compounds containing a carbamate group at C-5.

**Figure 32 molecules-28-06641-f032:**
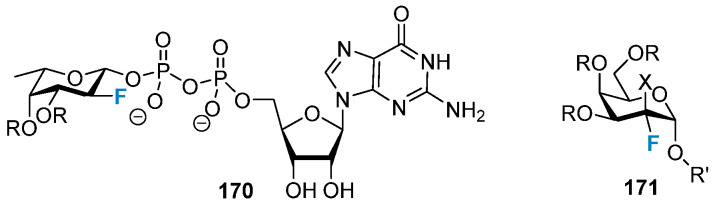
Fucosylation inhibitors and GDP-fucostatin galactose-1-phosphate uridyltransfer-ase inhibitors.

**Figure 33 molecules-28-06641-f033:**
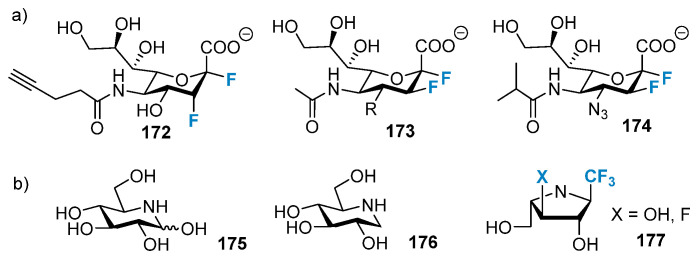
(**a**) 2,3-Difluorosialosides as covalent sialidase inhibitors. (**b**) Nojirimycin, 1-deoxynojirimycin, and fluorinated analogs of iminosugars.

**Figure 34 molecules-28-06641-f034:**
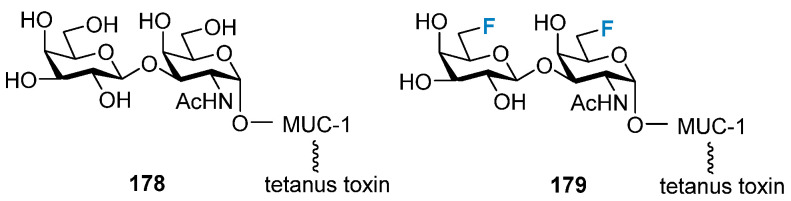
Fluorinated Thomsen–Friedenreich (TF) conjugates.

**Figure 35 molecules-28-06641-f035:**
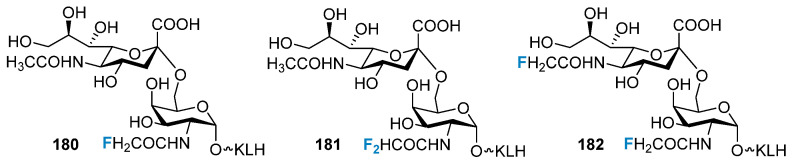
Fluorine-modified STn.

## Data Availability

Not applicable.

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
