# Peer review of "Drug Discovery Based on Fluorine-Containing Glycomimetics"

_molecules, 2023, doi:10.3390/molecules28186641_

Round 1
Reviewer 1 Report
The ms entitled: "Drug Discovery Based on Fluorine-Containing Glycomimetics" is an excellently written and thoroughly comprehensive review about the fluorination methods used to synthesize fluoro-sugars, the effects of fluorination on the physical properties of carbohydrates and their biological applications. This is a good quality work that is worth to be published in Molecules after minor revisions.
In particular:
1) Although the subject is very interesting it was previously reviewed in the following publication: Hevey R. The Role of Fluorine in Glycomimetic Drug Design. Chemistry. 2021; 27(7):2240-2253. doi: 10.1002/chem.202003135. This recent and similar review article should be cited by the authors.
2) References should be revised, the title of the journal is missing in some of them (i.e. 41a and 75)
Author Response
The ms entitled: "Drug Discovery Based on Fluorine-Containing Glycomimetics" is an excellently written and thoroughly comprehensive review about the fluorination methods used to synthesize fluoro-sugars, the effects of fluorination on the physical properties of carbohydrates and their biological applications. This is a good quality work that is worth to be published in Molecules after minor revisions.
We are deeply grateful for the comments provided by the reviewer regarding our manuscript. We highly value your concerns and have taken them into careful consideration. In the following section, we provide a point-by-point response to each of your comments. We highlighted all the revisions in yellow colour. We hope that our responses adequately address your concerns.
Comment 1:
1) Although the subject is very interesting it was previously reviewed in the following publication: Hevey R. The Role of Fluorine in Glycomimetic Drug Design. Chemistry. 2021; 27(7):2240-2253. doi: 10.1002/chem.202003135. This recent and similar review article should be cited by the authors.
Response:
We appreciate your valuable suggestion. The reference is very helpful for us to improve the integrity of our manuscript. We added the reference which is cited in [31] and other related reviews cited in [6], [7], [120], [121], respectively.
Comment 2:
References should be revised, the title of the journal is missing in some of them (i.e. 41a and 75).
Response:
We are grateful to the reviewer for bringing this matter to our attention. We have added the title of corresponding references, which are cited in [45a], [79], respectively.
Reviewer 2 Report
This manuscript presents an interesting overview of fluorine-containing glycomimetics. To my opinion, this review deserves to be published in Molecules after minor revisions.
Major points
The authors logically highlight the advantages of fluorine for the design of compounds of therapeutic interest. However, there are more and more concerns about fluoroalkyl substances due to the fact that these compounds and their degradation products may persist in the environment longer than any other man-made chemicals. Further concerns are their bioaccumulation and potential toxicological effects. It is proposed in the EU to ban many of fluorine-containing compounds (see for example: Org. Process Res. Dev. 2023, 27, 8, 1422–1426). I think that this point has to be discussed in this review (introduction or conclusion).
Page 18, line 605. The authors claim that many research groups have tried to synthesize fluoride-containing iminosugars without giving reference or examples of structures.
Page 6, Figure 9, the mechanistic arrows to depict the transformation of 58 are not correct. One arrow is missing (radical mechanism).
Minor points
Page 1, lines 36-42. These two references on glycomimetics should be cited : (a) Pharmaceuticals 2019, 12, 55. (b) Biomimetics 2019, 4, 53.
For all the sugars, for example D-glucose, the font size must be reduced for D or L in the text.
Page 1, line 42 repetition of “control” in the same sentence
Page 2, line 81. The authors should comment the interesting regioselectivity of reaction depicted in Figure 2a.
Page 4, line 32. Glycolide is a misleading term. Compound 32 is a lactone, glycolide is a bis-lactone, namely 1,4-dioxane-2,5-dione.
Page 6, line 186, Clostridium jejuni in italic
Page 9 and 10. The subtitle of this part “2.5 fluorination of carbohydrates by de novo strategy” is misleading. According to me, the synthesis of 110 and 116 are not de novo synthesis (107 and 111 are synthesized from sugars).
Page 9, Figure 19, the structure of 113 is not correct. According to me, at least three asymmetric centers (C2,C3,C4) have to be inverted.
Page 9, compound 117, Figure 20; The C-CF3 bond is not correctly depicted (117 is a planar molecule).
Page 13, line 383. This sentence has to be changed. X-ray diffraction studies give information on the solid-state conformation of molecules.
Page 13, line 421 structures and not tructures
Page 14, line 429-431. For clarity’s sake, please rephrase.
Page 14, line 435: For clarity’s sake, please rephrase. Do the authors mean exocyclic -O atom?
Page 15, Figure 27, For clarity’s sake, show the two protons at C-6 in 149-152
Page 15, Figure 28, For clarity’s sake, for 153-156, use also a relative rate of hydrolysis scale with 1 as the reference for 153 as done below.
Page 16, Figure 29, Not clear: is it k1/Ki? If not what is ki?
Page 16, line 535, reviews on glycosyltransferase inhibitors have to be cited: Bioorg. Med. Chem. 2001, 9, 3077; OBC, 2021, 19, 5439.
Page 18, line 588. Please define more precisely what is a O-5 or a C-1 transition state.
Page 18, line 603. The following books on iminosugars have to be cited: Iminosugars as Glycosidase Inhibitors: Nojirimycin and Beyond; Stütz, A. E., Ed., Wiley-VCH, New York; 1999. Iminosugars, from Synthesis to Therapeutic Applications; Compain, P., Martin, O. R., Eds., Wiley-VCH:Weinheim, 2007.
Page 19, lines 621-626. The discussion on the unique properties of fluorine has been presented before in the review. It sounds like an unnecessary repetition.
Page 30, ref 30 : J. Org. Chem instead of Org. Chem.
see above
Author Response
This manuscript presents an interesting overview of fluorine-containing glycomimetics. To my opinion, this review deserves to be published in Molecules after minor revisions.
We would like to express our sincere appreciation for the careful review of our manuscript and the insightful questions and suggestions raised. We highly value your concerns and have taken them into careful consideration. In the following section, we provide a point-by-point response to each of your comments. We highlighted all the revisions in yellow colour. We hope that our responses adequately address your concerns.
Comment 1:
The authors logically highlight the advantages of fluorine for the design of compounds of therapeutic interest. However, there are more and more concerns about fluoroalkyl substances due to the fact that these compounds and their degradation products may persist in the environment longer than any other man-made chemicals. Further concerns are their bioaccumulation and potential toxicological effects. It is proposed in the EU to ban many of fluorine-containing compounds (see for example: Org. Process Res. Dev. 2023, 27, 8, 1422–1426). I think that this point has to be discussed in this review (introduction or conclusion).
Response:
These are highly thought-provoking questions. We have stated the potential impacts of fluorine-containing compounds on environmental in the conclusion, while emphasizing the importance of fluorine-containing substitutes in pharmaceutical region.
Comment 2:
Page 18, line 605. The authors claim that many research groups have tried to synthesize fluoride-containing iminosugars without giving reference or examples of structures.
Response:
We greatly appreciate your suggestion. We have added reviews on the synthesis of fluorinated analogues of iminosugars, which are cited in [142-145]. We have also added the structure of polyhydroxylated pyrrolidines in Figure 33.
Comment 3:
Page 6, Figure 9, the mechanistic arrows to depict the transformation of 58 are not correct. One arrow is missing (radical mechanism)
Response:
We appreciate for pointing out the mistake. We have added the arrow showing the loss of a proton when a Ag(II) species oxidizes the primary hydroxyl group to give alkoxyl radical in Figure 9.
Comment 4:
Page 1, lines 36-42. These two references on glycomimetics should be cited : (a) Pharmaceuticals 2019, 12, 55. (b) Biomimetics 2019, 4, 53.
Response:
We appreciate for your valuable suggestion. The references are very helpful for us. We have added the references which are cited in [6] , [7].
Comment 5:
For all the sugars, for example D-glucose, the font size must be reduced for D or L in the text.
Response:
We appreciate you pointing out the mistake. We have reduced the font size of all the D or L of sugars in the manuscript.
Comment 6:
Page 1, line 42 repetition of “control” in the same sentence
Response:
We appreciate for your advice. We have rewritten the sentence as “it can serve as a considerable element to control molecular conformation” in the manuscript.
Comment 7:
Page 2, line 81. The authors should comment the interesting regioselectivity of reaction depicted in Figure 2a.
Response:
We appreciate for your suggestion. We have rewritten the sentence and explained the SN2 mechanism that the reaction undergoes.
Comment 8:
Page 4, line 32. Glycolide is a misleading term. Compound 32 is a lactone, glycolide is a bis-lactone, namely 1,4-dioxane-2,5-dione.
Response:
We appreciate you pointing out the mistake. We have replaced the name of 32 with lactone.
Comment 9:
Page 6, line 186, Clostridium jejuni in italic.
Response:
We appreciate you pointing out the mistake. We have changed the font of the Clostridium jejuni to italic.
Comment 10:
Page 9 and 10. The subtitle of this part “2.5 fluorination of carbohydrates by de novo strategy” is misleading. According to me, the synthesis of 110 and 116 are not de novo synthesis (107 and 111 are synthesized from sugars).
Response:
We appreciate for your comments. This subtitle “de novo strategy” is not accurate and we have changed it to “fluorine-containing building block strategy” which is more appropriate for the synthesis of 110 and 116. We have applied “fluorine-containing building block strategy” in the context of 2.5 .
Comment 11:
Page 9, Figure 19, the structure of 113 is not correct. According to me, at least three asymmetric centers (C2,C3,C4) have to be inverted.
Response:
We appreciate you pointing out the mistake. We have inverted the asymmetric centers of C2, C3, C4 as your suggestion.
Comment 12:
Page 9, compound 117, Figure 20; The C-CF3 bond is not correctly depicted (117 is a planar molecule).
Response:
We appreciate you pointing out the mistake. We have corrected the structure of 117 .
Reviewer 2, comment 13:
Page 13, line 383. This sentence has to be changed. X-ray diffraction studies give information on the solid-state conformation of molecules.
Response:
We appreciate for your suggestion. We have removed “solution-state” to make the sentence compatible with X-ray diffraction.
Comment 14:
Page 13, line 421 structures and not tructures
Response:
We appreciate you pointing out the mistake. We have corrected the word “structures”.
Comment 15:
Page 14, line 429-431. For clarity’s sake, please rephrase.
Response:
We appreciate for your suggestion. We have rephrased this part as “ In carbasugars, C1–C2 and C1–C5a have the same polarity, therefore no preferential conformation is observed. Leclerc et al. proposed a CF2 analogue of carbasugars in which the CF2 group replaced the endocyclic oxygen atom to restore the exo conformation attributable to a polarization of the C1–CF2 bond [96] (Figure 26)” and added explanation of “exo-anomeric effect and exo conformation” in line 419-422. We have explained the specificity of carbasugars and CF2 modification facilitates restoring of exo conformation, which shall make it easier to understand.
Comment 16:
Page 14, line 435: For clarity’s sake, please rephrase. Do the authors mean exocyclic -O atom?
Response:
We appreciate for your suggestion. We have rephrased this part as “Studies have shown that the replacement of the endocyclic oxygen atom by a CF2 group in maltose restored stereoelectronic stabilization caused by the lone pairs of electrons on the anomeric oxygen and the σ*C1-O1 orbital. The replacement resulted in the preferential generation of the exo conformation [97]”. We have explained the contribution of a CF2 group in the exo conformation caused by anomeric oxygen, which shall make it easier to understand.
Comment 17:
Page 15, Figure 27, For clarity’s sake, show the two protons at C-6 in 149-152
Response:
We appreciate for your suggestion. We have showed the two protons at C-6 of 149-152 in Figure 27.
Comment 18:
Page 15, Figure 28, For clarity’s sake, for 153-156, use also a relative rate of hydrolysis scale with 1 as the reference for 153 as done below.
Response:
We appreciate for your suggestion. We have used rate of hydrolysis of 157 as 1 in Figure 28.
Comment 19:
Page 16, Figure 29, Not clear: is it k1/Ki? If not what is ki?
Response:
We appreciate for your suggestion. We failed to find the definition of ki in the reference or literature that directly compare the hydrolysis rates between 162 and 163. We have removed the data “k1/Ki” in Figure 29 for the sake of accuracy and preciseness of data.
Comment 20:
Page 16, line 535, reviews on glycosyltransferase inhibitors have to be cited: Bioorg. Med. Chem. 2001, 9, 3077; OBC, 2021, 19, 5439.
Response:
We appreciate for your suggestion. The references are very helpful for us. We have added the reference which are cited in [120], [121].
Comment 21:
Page 18, line 588. Please define more precisely what is a O-5 or a C-1 transition state.
Response:
We appreciate for your suggestion. We have replaced the statement with “oxacarbenium ion-like transition states”, which is more concise and acknowledgeable.
Comment 22:
Page 18, line 603. The following books on iminosugars have to be cited: Iminosugars as Glycosidase Inhibitors: Nojirimycin and Beyond; Stütz, A. E., Ed., Wiley-VCH, New York; 1999. Iminosugars, from Synthesis to Therapeutic Applications; Compain, P., Martin, O. R., Eds., Wiley-VCH:Weinheim, 2007.
Response:
We appreciate for your suggestion. The books on iminosugars enrich our manuscript. We have cited them in [144], [145].
Comment 23:
Page 19, lines 621-626. The discussion on the unique properties of fluorine has been presented before in the review. It sounds like an unnecessary repetition.
Response:
We appreciate for your suggestion. We have removed the corresponding content from the article.
Comment 24:
Page 30, ref 30 : J. Org. Chem instead of Org. Chem.
Response:
We appreciate you pointing out the mistake. We have corrected the formatting of [34] and rechecked the formatting of all the references.